# The SMN Complex at the Crossroad between RNA Metabolism and Neurodegeneration

**DOI:** 10.3390/ijms24032247

**Published:** 2023-01-23

**Authors:** Irene Faravelli, Giulietta M. Riboldi, Paola Rinchetti, Francesco Lotti

**Affiliations:** 1Department of Stem Cell & Regenerative Biology, Harvard University, Cambridge, MA 02138, USA; 2Center for Motor Neuron Biology and Diseases, Departments of Pathology & Cell Biology, and Neurology, Columbia University Irving Medical Center, New York, NY 10032, USA; 3The Marlene and Paolo Fresco Institute for Parkinson’s and Movement Disorders, NYU Langone Health, New York, NY 10017, USA

**Keywords:** spinal muscular atrophy, RNA metabolism, survival motor neuron, SMN complex, neurodegeneration, mRNA splicing

## Abstract

In the cell, RNA exists and functions in a complex with RNA binding proteins (RBPs) that regulate each step of the RNA life cycle from transcription to degradation. Central to this regulation is the role of several molecular chaperones that ensure the correct interactions between RNA and proteins, while aiding the biogenesis of large RNA-protein complexes (ribonucleoproteins or RNPs). Accurate formation of RNPs is fundamentally important to cellular development and function, and its impairment often leads to disease. The survival motor neuron (SMN) protein exemplifies this biological paradigm. SMN is part of a multi-protein complex essential for the biogenesis of various RNPs that function in RNA metabolism. Mutations leading to SMN deficiency cause the neurodegenerative disease spinal muscular atrophy (SMA). A fundamental question in SMA biology is how selective motor system dysfunction results from reduced levels of the ubiquitously expressed SMN protein. Recent clarification of the central role of the SMN complex in RNA metabolism and a thorough characterization of animal models of SMA have significantly advanced our knowledge of the molecular basis of the disease. Here we review the expanding role of SMN in the regulation of gene expression through its multiple functions in RNP biogenesis. We discuss developments in our understanding of SMN activity as a molecular chaperone of RNPs and how disruption of SMN-dependent RNA pathways can contribute to the SMA phenotype.

## 1. Introduction

Survival Motor Neuron (SMN) is a ubiquitously expressed protein composed of 294 amino acids with multiple domains that provides a platform for RNA and protein binding during its function as molecular chaperone of ribonucleoprotein (RNP) complexes [1,2,3]. SMN interacts with itself and other proteins to form the SMN complex, a multi-protein complex essential for the assembly of small nuclear RNPs that consist of uridine (U) rich non-coding RNAs bound by selected proteins [4]. These RNPs are crucial for the regulation of pre-mRNA splicing and histone pre-mRNA 3′-end processing [5]. In addition, SMN has been implicated in the formation of several other cellular RNPs containing both coding and non-coding RNAs especially those involved in axonal mRNA trafficking and translation [6,7].

SMN is encoded by the *survival motor neuron 1* (*SMN1*) gene located on chromosome 5q13 [8]. Humans also carry a paralogous gene (referred as *SMN2*) that differs from *SMN1* by a small pool of nucleotides, with the most significant difference being a C-T transition in exon 7. This modification creates a splicing alteration that leads to skipping of exon 7 (SMN∆7) in approximately 90% of SMN2 transcripts, which are therefore unable to generate a full-length SMN protein [1]. Thus, only approximately 10% of SMN2 transcripts are translated into a functional protein, while the remaining transcripts generate a highly unstable protein that is rapidly degraded [9].

Mutations or deletions in the SMN1 gene result in spinal muscular atrophy (SMA), a neuromuscular disorder with onset mainly during the early childhood. With a prevalence ranging from 8 to 10 per 100,000 individuals and an incidence of 1:10,000 live births, SMA represents the leading cause of infant death due to a monogenic disease [2,10,11]. SMA presents with a variable clinical phenotype with four major subtypes based on severity, ranging from type 0 (prenatal onset) to type 4 (adulthood onset) at either extreme of the spectrum [11]. The most common presentation is SMA type 1, affecting more than 60% of all SMA patients. Children with SMA type 1 are unable to sit, maintain head control, or eat independently. In the natural history of the disease, they require early mechanical ventilation, with mortality usually caused by respiratory failure [12,13].

The clinical phenotype directly correlates with the number of SMN2 copies present in the genome. Patients with more than three copies of SMN2 usually display milder SMA symptoms, confirming that SMN2 functions as a compensatory mechanism and further associates the loss of fully functional SMN with the SMA phenotype [13,14]. Decades of in vitro and in vivo studies have investigated the genetic and functional role of SMN in physiological conditions and in the context of SMA. The SMA phenotype is most likely the result of deficits in one or multiple SMN-dependent RNA pathways, whose impairment contribute either independently or synergistically to disease etiology. In line with this hypothesis, a combination of alterations due to the lack of the SMN complex activity might contribute to selected aspects of SMA pathology. A full elucidation of the multifaceted SMN biology is crucial in order to understand the plethora of functions of the complex, their connection with the disease and finally provide complementary and/or optimized approaches to treat all SMA forms.

In this review, we focus our attention on the pleomorphic role of the SMN complex in RNA metabolism and processing, first examining its various functions and then detailing how their dysfunctions result in the SMA phenotype. We provide a comprehensive overview of canonical and recently discovered functions of the SMN complex, covering basic molecular concepts and adding a translational perspective.

## 2. SMN Complex Function in RNA Metabolism

The evolutionarily conserved and ubiquitously expressed SMN protein localizes to both the cytoplasm and the nucleus where it accumulates in nuclear structures known as Gems (Gemini of Cajal bodies). The observation that Gems interact with Cajal bodies (CBs), rich in p80/Coilin and RNAs, provided the first indication that SMN functions in RNA metabolism [15]. To this end, SMN functions through self-oligomerization and through the interaction with other proteins (i.e., Gemins2-8 and Unrip) to form a large multi-protein complex—the SMN complex (for detailed characterization see [1,3]).

SMN highly conserved domains include a short segment near the N-terminus that is responsible for binding with high affinity to Gemin2 [16]; the Tudor domain that recognizes symmetric dimethylarginine modifications in arginine/glycine (RGG) rich regions in a number of proteins involved in RNA processing, including the Sm proteins [17,18]; and the evolutionarily conserved C-terminal YG-box that is required for SMN oligomerization and is impaired in carriers of some SMN variants that lead to SMA [19,20]. Structural studies demonstrated that the SMN YG-box forms helical oligomers mediated by glycine zippers [21].

Several of these integral components are found in distinct complexes outside of the SMN complex itself (i.e., Gemin3/4, Gemin6/7/Unrip, and Gemin5 either alone or associated with Gemin3/4) [22,23,24,25,26,27]. Among these SMN-interacting components, Gemin5 is crucial for regulating the expression of SMN itself. Indeed, Gemin5 activates SMN translation and Gemin5′s mRNA-binding activity is regulated by SMN levels, in an interesting feedback loop [26].

The SMN complex has been showed to assemble in a stepwise manner through multiple modular additions. These findings also reflect an expansion in the complexity of the SMN complex throughout evolution. Accordingly, the multi-protein human SMN complex arose from an ancestral version in fission yeast comprised of solely SMN and Gemin2 that underwent addition of multiple Gemin proteins [28]. Importantly, SMN and all Gemins are essential for viability in all organisms from yeast to mammals, indicating that they perform essential cellular functions [29].

On the other side, some of the components of the SMN complex also act independently to modulate the protein synthesis. This is the case of Gemin 5. Gemin 5 plays a crucial role in RNA translation both in synergy with SMN, through binding and delivering RNAs to the SMN complex, as well as outside the SMN complex [24,25,26]. Gemin5 interacts with the ribosomes and regulates translation either activating or repressing it through the involvement of different domains [30,31,32].

In the following section, we provide a wide overview of the roles played by the SMN complex in RNA processing, describing both the well-established pathways and the most recently described ones (Figure 1).

## 3. Spliceosomal snRNP Assembly

Precursor messenger RNA (pre-mRNA) splicing is a key process during post-transcriptional regulation of gene expression, as it enables the production of a large variety of proteins from a limited pool of genes. Up to 95% of human genes undergo alternative splicing with selective removal of specific introns by spliceosome RNP-dependent machinery [33]. The SMN complex is fundamental for the assembly of spliceosomal small nuclear RNPs (snRNPs) that function in pre-mRNAs splicing [34,35]. The major spliceosome (U2-dependent) is responsible for the splicing of the vast majority of introns and is comprised of U1, U2, U4/U6, and U5 snRNPs [36]. The minor spliceosome (U12-dependent) is responsible for the processing of a small number of introns (U12 introns) and is comprised of U11, U12, U4atac/U6atac, and U5 snRNPs [37]. Each spliceosomal snRNP consists of one uridine-rich small nuclear RNA, a heptameric ring of Sm proteins (B/B′, D1, D2, D3, E, F, and G) and a specific set of accessory proteins. The biogenesis of spliceosomal snRNPs is a multistep process that requires the SMN complex and both nuclear and cytoplasmic phases (Figure 1).

In the cytoplasm, the different Sm proteins sequentially associate with each other and with chloride conductance regulatory protein (pICln), which recruits all newly synthesized Sm proteins to the protein arginine methyltransferase 5 (PRMT5) complex [38,39]. The PRMT5 complex consists of the methyltransferase PRMT5, the assembly chaperone pICln, and WD45 (also known as MEP50). The main role of the PRMT5 complex is to catalyze the symmetrical arginine-dimethylation in SmB/B′, SmD1, and SmD3 and the formation of higher-order Sm protein complexes, which are then released and transferred to the SMN complex for the late phase of the assembly [34,40,41]. One of the Gemin proteins, Gemin 2, also plays a crucial role in Sm selection and snRNP assembly, as well as in the release of the SMN complex from the nucleus to the cytoplasm [42]. Concurrently, the snRNAs are transcribed in the nucleus and then transferred to the cytoplasm through the interaction with shuttling proteins, e.g., phosphorylated adaptor for RNA export (PHAX), Cap-binding complex (CBC), exportin (Xpo1) or ras-related nuclear protein GTP (Ran) [43]. In the cytoplasm, the SMN complex interacts with U snRNAs to promote their assembly with a toroidal ring of Sm proteins. The Sm ring is a common structural denominator of the U snRNPs that enables several subsequent steps in the biogenesis pathway. This includes the conversion of the m7G-cap of the U snRNAs to its hypermethylated form and the nuclear import of the assembled U snRNP particle [44,45,46,47]. Once again in the nucleus, the SMN complex is localized with the snRNPs and other proteins, such as Coilin, in the Cajal bodies (CBs), where later maturation of snRNP occurs [48].

Cajal body condensation is mediated by different prost-translational modification (PTM) events. These include mTOR-dependent phosphorylation of serine 49 and 63 on SMN, arginine methylation, acetylation, as well as sumoylation [49,50,51,52]. Thus, SMN acts as a molecular chaperone of RNPs by preventing promiscuous bindings and providing efficiency and specificity to the process of spliceosomal snRNP assembly [34,53].

## 4. U7 snRNP Assembly

The U7 snRNP is the only example of a U snRNP that has no function in mRNA splicing but is exclusively dedicated to the 3′-end processing of a select subset of mRNAs that code for the replication-dependent histone proteins [54,55]. The metazoan replication-dependent histone genes encode the only eukaryotic cellular mRNAs that are not polyadenylated, ending instead in a conserved stem-loop sequence [56,57]. The replication-dependent histone genes are present in a specialized nuclear domain, the histone locus body (HLB). The HLB creates an optimal environment for efficient processing by concentrating factors necessary for histone mRNA biogenesis [58]. U7 snRNP facilitates 3′-end formation of histone mRNAs by directly associating with the mRNA through anti-sense base pairing of U7 snRNA and facilitating the recruitment of factors that carry out the cleavage reaction [57,59].

The SMN complex is required for the assembly of the U7 snRNP [55] (Figure 2). While U7 and spliceosomal snRNPs follow the same biogenesis pathway, the U7 snRNA contains a unique mixed Sm/LSm core comprising Sm proteins B, D3, E, F, G, and two LSm proteins (LSm10 and LSm11) that replace SmD1 and SmD2 [55,60]. SMN-dependent disruption of U7 snRNP assembly leads to a severe reduction in the levels of mature U7 and concomitant accumulation of a previously uncharacterized U7 pre-snRNA [61]. However, the process by which the SMN complex is able to distinguish U7 snRNA targets that receive Sm/LSm cores from splicesomal snRNAs that receive the canonical Sm core is still unknown. Certainly, a mechanism for discriminating between the two classes must exist, as the assembly of RNAs with improper Sm cores would be highly detrimental to cells. It is well established that the Gemin5 subunit of the SMN complex is responsible for identifying snRNAs and delivering them to the SMN complex for assembly [1]. Early evidence suggests that Gemin5 does not bind U7 snRNA [1] leaving open the intriguing possibility that a separate and yet to be identified factor is responsible for specifically recruiting U7 snRNA to the SMN complex competent for U7 assembly.

The functional consequence of U7 snRNP reduction caused by SMN deficiency is a disruption of replication-dependent histone mRNA 3′-end processing resulting in accumulation of aberrantly poly-adenylated transcripts. This indicates that the SMN-dependent reduction in functional U7 snRNP levels does not simply reduce the rate of processing histone transcripts, but switches processing to an entirely different modality of 3′-end processing. The loss of this unique histone mRNA 3′-end structure ultimately leads to dysregulated histone gene expression [61].

## 5. Messenger RNP Assembly and Axonal Transport

Despite the well-characterized role of SMN in the assembly of U snRNPs, there are numerous lines of evidence that situate SMN in the center of a wide variety of RNA processing pathways through interactions with various RNA-binding proteins (RBPs) (Figure 3). The vast majority of these RBPs have critical roles in different aspects of post-transcriptional regulation of gene expression [62,63,64,65,66], including mRNA transport, stability and local translation in neurons. SMN binds to RBPs in vitro through the RG-rich domains in a methylation-dependent manner. In cultured neurons, SMN localizes within axonal and dendritic granules along with mRNA-binding proteins such as hnRNP-R, HuD, KSRP and IMP1 [67,68]. These granules exhibit dynamic movement along the axon and contain other integral components of the SMN complex but not Sm proteins [63,64,69,70], suggesting that SMN also functions outside of snRNP assembly. Importantly, SMN deficiency causes decreased localization of RBPs and their associated mRNAs in axons and growth cones of developing neurons in vitro [62,64,69]. Local translation of mRNAs is essential for proper neuronal pathfinding and, accordingly, these SMN-deficient neurons display reduced neurite length and smaller growth cones [62,64,69]. Collectively, these findings suggest that SMN interaction with various RBPs and their associated mRNA targets may contribute to neuronal mRNA trafficking, perhaps by facilitating the interaction of RBPs with their mRNA targets mimicking its role in snRNP assembly.

Interestingly, SMN also associates with RBPs involved in other aspects of mRNA regulation and with hundreds of transcripts containing AU-rich elements (AREs) in their 3′ UTR [71] that are key for the regulation of their turnover [71]. One well-described example is Cdkn1a mRNA, which is antagonistically regulated by both KSRP and HuD, whose mRNA levels increase upon SMN deficiency due to increased stability [65,66]. Therefore, SMN may additionally function in the cytoplasmic turnover of AU-rich transcripts by mediating their association with specific RBPs. However, the molecular function(s) of SMN in the biology of mRNPs and other candidate RNP targets is unknown and awaits the development of specific assays to provide mechanistic insights into the full spectrum of SMN-mediated RNA regulation.

## 6. R-Loop Resolution and Transcription Termination

R-loops consist of RNA/DNA hybrids, formed during transcription when nascent RNA hybridizes to the DNA template strand, displacing the non-template DNA strand [72]. They were first described in 1976 [73], but they could be only investigated in recent years thanks to scientific advances, such as the development of specific antibodies to target R-loops (S9.6) and DNA-RNA immunoprecipitation coupled to deep sequencing (DRIP-Seq) [74,75,76]. The picture emerging from these studies suggests that R-loops can be both beneficial and deleterious to cells. Paradoxically, while they are required for important biological processes, they can also promote DNA damage and genome instability [77]. Mutations that have been found to cause R-loop formation include those causing impairment in transcript generation, mRNA splicing and processing, and RNA degradation [78,79]. Altogether, these findings imply that molecules with a role in RNA metabolism could be crucial for R-loop biogenesis.

In line with the fact that R-loops can cause genomic instability, factors implied in DNA damage response are also able to prevent R-loop formation or solve them once formed. They include three topoisomerases (I, II, and IIIB) and RNA helicases [80,81]. In particular, human Senataxin (SETX) can process R-loops at transcription termination sites and lack of SETX results in R-loop increase and impairment of meiosis in the animal model [82]. In the clinical setting, SETX mutations were documented in a juvenile form of amyotrophic lateral sclerosis [83] and in patients with severe early onset ataxia with oculomotor apraxia (AOA2) [84].

A recent study investigated the mechanism of methylation of RNAP II carboxy-terminal domain (CTD) [85], dependent on PRMT5. The methylated form of RNAP II CTD binds SMN, which in turn interacts with SETX and recruits it at the methylation site (Figure 4). Loss-of-function of any of these factors increases R-loop accumulation at the transcriptional termination region, suggesting that SMN, along with the other proteins, contribute to R-loop resolution during transcription termination. Based on their observations, the authors propose that genome instability caused by less efficient removal of R-loops may contribute to neurodegeneration. However, the pathways driving motor neuron loss downstream of R-loop accumulation are not fully understood.

## 7. RNA Metabolism Defects in SMA

SMA is the leading genetic cause of infant death and the second most common autosomal recessive genetic disorder after cystic fibrosis [11,86]. SMA causes degeneration and loss of alpha motor neurons in the anterior horn of the spinal cord leading to progressive muscle weakness and, in severe cases, respiratory failure and death, with differential vulnerability of distinct motor neuron pools [87]. Some preservation of muscle power in milder cases of SMA can be associated with collateral axonal sprouting and myofiber re-innervation, supported by the presence of large motor units and fiber type grouping [87].

The development of various animal models of SMA has been valuable to defining the cellular and molecular basis of the disease [29,88]. Complete absence of SMN in model organisms is lethal as it likely is in humans, where the absence of both SMN1 and SMN2 has never been observed. In contrast, ubiquitous SMN reduction to levels found in severe SMA patients appears to be tolerated relatively well by most tissues, except the nervous system [29]. Thus far, the genetic and phenotypic hallmarks of the human disease have been accurately reproduced in mouse models, which have proven to be a powerful platform for elucidating pathological mechanisms in SMA [29,89,90,91].

Examination of cellular pathology reveals significant loss of motor neurons in the spinal cord of SMA mice [90,91,92,93]. SMA motor neurons display differential, segment-specific vulnerabilities along the rostro-caudal axis [93]. Beyond death of spinal motor neurons, SMA pathology is characterized by additional defects that occur both centrally, at synapses impinging on somata and dendrites of motor neurons, and distally, at the neuromuscular junction (NMJ). The NMJ defects include presynaptic neurofilament accumulation, reduced vesicle content, and impaired synaptic transmission as well as defective postsynaptic acetylcholine receptor clustering and motor endplate development [51,94,95,96,97,98,99]. In addition to muscles, motor neurons wire together with other neurons into functional circuits essential for motor behavior. Disruptions within these neural networks likely contributes to SMA, as SMA motor neurons receive fewer glutamatergic synaptic inputs from proprioceptive sensory neurons and local spinal interneurons [93,97]. SMA mice indeed demonstrate severe impairment of sensory-motor neurotransmission associated with dysfunction of proprioceptive synapses [93,100].

In sum, SMA is emerging as a disease characterized by dysfunction of multiple components of the motor system and not only dysfunctioning motor neurons. Defects in heart, pancreas, and liver have been documented in severe SMA mice [101,102,103,104,105], however the specific dysfunction of peripheral tissue in human SMA remains to be identified [106,107].

In the following section, we provide a broad outline of how dysfunctions in selective SMN-dependent RNA pathways contribute to the SMA phenotype (Figure 1).

## 8. Spliceosomal snRNP Assembly Defects and Its Consequences on mRNA Splicing

The proper assembly of the spliceosome machinery relies on the chaperone function provided by the SMN complex. Altered snRNP production due to SMN deficiency can lead to the neuromuscular defects that are hallmarks of the SMA phenotype (Figure 1).

Biochemical quantification of snRNP assembly in cellular and animal models of SMA showed a strong correlation between SMN expression levels and disease severity. Accordingly, both fibroblasts and lymphoblasts derived from SMA Type I patients have a reduced capacity for assembly of snRNPs [108]. Delivery of mature snRNPs lacking SMN is sufficient to rescue SMA phenotypes in SMN-deficient zebrafish embryos [109]. The disease phenotype was similarly rescued in mice following the introduction of the SMNA111G allele, which retains its snRNP assembly function [110]. Interestingly, reduced expression of pICln—a component of the PRMT5 complex required in the early steps of snRNP assembly—causes SMA-like defects in zebrafish and Drosophila [109,111]. Moreover, alterations in the steady-state levels of snRNPs have been reported in tissues from SMA mice and are more prominent in spinal cord and brain extracts, compared to other tissues, where snRNP levels were not decreased [110,112,113].

Importantly, the reduction in steady-state snRNP levels is not only moderate but also differential. In spinal cords from SMA mice, the components of the U12-dependent spliceosome are more significantly reduced, as compared to other tissues from the same animals [110,112,113]. Defects in SMN were found to perturb the splicing of several genes that contain U12 introns in both human SMA lymphoblasts and SMN-deficient mouse spinal cord [114,115,116,117]. Furthermore, SMN deficiency provokes a much more prominent reduction of snRNPs in motor neurons relative to other neurons in the spinal cord of SMA mice [118]. This has been attributed to both the intrinsic low degree of exon 7 inclusion in SMN2 transcripts within normal motor neurons and the activation of a negative feedback loop in which reduced snRNPs further decrease exon 7 splicing in SMA [118,119]. These observed qualitative and quantitative differences in snRNP amount and rate of assembly across SMA tissues provides an explanation of how a dysfunctional SMN pathway might lead to defective RNA processing within the motor system.

While there is little doubt that SMN deficiency causes splicing alterations, the downstream consequences and contribution of this dysfunction to the SMA phenotype is less elucidated. One possibility is that widespread splicing alterations cause the SMA phenotype. This scenario is founded on the identification of widespread splicing defects in tissues from SMA animal models [113,115,116,120]. Theories that widespread splicing alterations may cause selective motor neuron death in SMA are supported by mechanistic insights from recent work finding pervasive splicing defects, global DNA damage, and an activated stress response downstream of SMN-deficiency [116]. Alternatively, cell-type specific transcriptome and splicing profiles have identified several genes whose expression is selectively disrupted in motor neurons of SMA mice [121,122,123]. One study showed alteration in the expression of genes relevant for synaptic function and circuit formation (C1q, Er81) and NMJ maintenance (Agrin) [123].

Building on the observations that components of the minor spliceosome are preferentially reduced in the spinal cord of SMA mice and that this pathway may have a prominent role in neuronal homeostasis, a study analyzed all Drosophila minor intron-containing genes to identify those that are not only aberrantly spliced but also lead to functional defects in the Drosophila motor system [117]. SMN deficiency altered both splicing and expression of several minor intron-containing genes, including stasimon, whose expression and splicing was also found to be perturbed in the constituent neurons of the sensory-motor circuit in SMA mice. Restoring stasimon expression in the motor circuit was found to correct defects in NMJ transmission and muscle growth in Drosophila SMN mutants, as well as correct motor axon outgrowth and development defects in SMN-deficient zebrafish [117]. Correcting stasimon expression, however, was unable to fully rectify all defects caused by SMN deficiency, further indicating that the SMA phenotype is not the result of a single gene’s disruption but rather the result of multiple defects likely in distinct SMN-dependent pathways [117]. Nonetheless, the identification of Stasimon provided the first link between splicing perturbations in the motor circuit to phenotypic consequences of SMN deficiency in animal models, establishing a mechanistic framework explaining how neuronal selectivity can emerge from disruption of SMN general role in splicing.

A recent work looked at the specific role of U12 splicing alterations in different SMA mouse models of varying disease severity [124]. The authors selectively restored the U12 pathway through the delivery of minor snRNA genes targeting U12 splicing dysfunction. As a result, treated SMA mice showed a significant rescue of the loss of proprioceptive inputs on SMA motor neurons, which constitutes an early phenotype of the SMA circuit. In parallel, the animals exhibited a moderate improvement in the neuromuscular function. Altogether, these data reinforce the mechanistic link between SMN loss, U12 splicing alterations and synaptic impairment in SMA motor circuit.

Other studies have since focused on additional genes that are mis-spliced in SMA models including Neurexin2a (Nrxn2a) and Chondrolectin (Chodl), both of which are important for motor neuron axon outgrowth [125,126].

These studies demonstrate that SMN-dependent alteration of the snRNP biogenesis pathway can produce specific deficits that, when combined, result in profound motor dysfunction. Continued research is needed to identify the specific molecular mechanisms linking SMN deficiency to downstream target genes alteration and, secondarily, those genes’ relative contributions to the SMA phenotype in a mouse model of the disease.

## 9. U7 snRNP Assembly and Histone mRNA Processing Impairment

Consistent with the essential role of SMN for U7 snRNP biogenesis, the steady-state levels of U7 snRNP are strongly reduced in both SMN-deficient mammalian cells and tissues of SMA mice [61]. The reduction in U7 snRNP levels results in defective histone mRNA processing as evidenced by the accumulation of uncleaved, polyadenylated histone mRNA in tissues from SMA mice as well as postmortem tissue from SMA patients [61] (Figure 2). Importantly, SMN deficiency causes disruption of histone mRNA 3′-end formation in vulnerable SMA motor neurons, demonstrating the impairment of this RNA pathway in disease-relevant, post-mitotic cells. Histones play essential roles in epigenetic regulation of gene expression through the facilitation of dynamic changes in the structural organization of chromatin. Thus, even subtle alterations in the cellular pool of histones induced by SMN deficiency could have significant effects on gene regulation and detrimental consequences in SMA neurons.

Due to increased susceptibility of genotoxic stress in SMA patient-derived fibroblasts, SMN deficiency can reduce expression of histone H2AX and, consequently, decrease levels of its phosphorylated form during DNA damage [127]. The phosphorylation of H2AX is a critical initiating event during a cell’s response to DNA damage [128]. Mice lacking H2AX exhibit chromosomal instability, defects in the DNA repair process, and an inability of cells to recruit key DNA repair proteins to sites of DNA damage [129]. Thus, SMA cells may be more vulnerable to DNA damage if the histone factors that normally facilitate the DNA damage response are disrupted.

A recent study demonstrated that SMN-dependent assembly of U7 snRNP is required for NMJ integrity. The expression of U7-specific Lsm10 and Lsm11 proteins in SMA mice was able to enhance U7 snRNP assembly, rescue histone mRNA processing defects and finally ameliorate the neuromuscular phenotype [130]. Interestingly, this work showed that the absence of SMN causes impaired U7 snRNP biogenesis, which results in deregulation of histone gene expression ultimately leading to Agrin downregulation within motor neurons. Agrin has a crucial role in NMJ function and has been previously associated with SMA pathology; Agrin upregulation in the SMA mouse model was able to improve muscle fibers size and muscular innervation [131]. 

Altogether, these findings provided a compelling example of a molecularly defined cascade of events linking SMN deficiency with dysregulation of a select RNA processing pathway through the disruption of a specific RNP assembly activity carried out by SMN.

## 10. Altered mRNP Assembly and Axonal Transport

The association of SMN with multiple RBPs suggests it functions to modulate mRNA metabolism beyond nuclear processing by snRNPs. SMN-interacting RBPs are known to bind to a variety of cytoplasmic mRNAs and play a role in multiple regulatory aspects of their life cycle such as stability, transport, and translation. Dysregulation of any of these processes induced by SMN deficiency may contribute to neuronal dysfunction and thus be relevant to disease etiology (Figure 3). Of particular interest is SMN’s involvement during the assembly of messenger ribonucleoproteins (mRNPs) transported along axons for regulated local translation at the distal end of highly polarized cells such as neurons. Both β-actin mRNA and protein levels are reduced in distal axons and growth cones of primary motor neurons derived from severe SMA mice, which also show reduced axon length and smaller growth cones in culture [132].

The assembly and transport of mRNPs along the axon is crucial for proper development of the axonal terminals (for review of the role of SMN in this process see [7]). For example, SMN-associated RBPs are known to bind β-actin mRNA, and axons from cultured SMA motor neurons exhibit reduced length and smaller growth cones [63,69]. SMN is additionally important for the assembly of β-actin mRNA with Igf2-mRNA binding protein 1 (IMP1) [133]. Reduced levels of SMN are associated with smaller IMP1 granules, which are less effective at binding with ß-actin mRNA, impairing its localization along the axon [133]. However, β-actin KO in mouse motor neurons exerted no effect on axonal spreading and NMJ innervation, suggesting that other factors also play a role in the SMA axon phenotype. SMN is likely able to affect the assembly of a broad spectrum of mRNPs through the interaction with poly(A) binding proteins (PABP), and therefore affecting other targets as well [133]. In this role, SMN seems to function as a chaperone and a hub facilitating the assembly and folding of different proteins in more specific complexes.

Additionally, SMN has been reported to bind to HuD, a RBP important for mRNA stability during neural development [64,65]. The SMN-HuD complex interacts with the candidate plasticity-related gene 15 (cpg15) mRNA before being transported in the growth cones [62]. This complex appears to be involved in non-cell-autonomous regulation of dendritic and axonal growth. These results suggest that by interacting with RBPs SMN is responsible for processing axonal mRNAs and its deficiency can cause a depletion of proteins localized within the growth cones [134]. In particular, the cytoskeleton-associated growth-associated protein 43 (GAP43) mRNA and protein levels were reduced in SMA MNs derived from mouse embryos. This reduction in GAP43 mRNA and protein was rescued after overexpression of HuD and IMP1 and was accompanied by improved axonal morphology in SMA neuronal cultures [134]. More recently, Hao and colleagues further confirmed in vivo the importance played by the complex HuD-SMN in the axonal development using HuD mutants of zebrafish [135].

By further identifying SMN’s role in supporting and facilitating axonal development, we can begin to unravel how deficits in this complex contribute to neurodegeneration. Indeed, impaired assembly of mRNPs has been identified as a key mechanism of many neurodegenerative diseases, such as amyotrophic lateral sclerosis (ALS), in which stress granules containing mRNPs are hallmarks of disease pathology [136]. It seems probable that SMN supports chaperoning and assembly of axonal mRNPs rather than directly transporting mRNA along axons, similarly to its role as a chaperone during the biogenesis of snRNPs.

## 11. R-Loop Accumulation and Genomic Instability

Mutations in proteins that function in R-loop resolution are responsible for devastating—often neurodegenerative—human diseases and indicate that R-loops warrant further understanding in disease conditions [77]. Mutations in the putative RNA/DNA helicase SETX cause neurodegenerative diseases, the dominant juvenile form of amyotrophic lateral sclerosis type 4 (ALS4), and a recessive form of ataxia oculomotor apraxia type 2 (AOA2) characterized by progressive degeneration of motor neurons in the brain and spinal cord, muscle weakness and atrophy [84,137].

A recent study proposed that SMN controls R-loop resolution by interacting with the C-terminal domain of RNA polII and SETX [85] (Figure 4). However, the link between SMN function in R-loop resolution and motor neuron degeneration in SMA has not been established. Transcriptomic analysis of spinal cords from mice in an antisense oligonucleotide (ASO)-inducible model of SMA via SMN depletion showed a widespread intron retention, particularly of minor U12 introns [116]. Intron retention was concomitant with a strong induction of the p53 pathway and DNA damage response, manifesting as γ-H2A.X positivity in neurons of the spinal cord and brain. After SMN depletion in both human neuroblastoma cells and induced pluripotent stem cell-derived motor neurons, retained introns were enriched in R-loops compared with spliced introns. Moreover, hallmarks of DNA breaks and damage could be detected in SMA mice and in cell cultures. Overexpression of RNase H1, which is able to solve the R-loops, could partially rescue the DNA damage in vitro. Based on these data, Jangi et al. propose that significant SMN reduction triggers widespread splicing defects, which promote global DNA damage through the generation of R-loops specifically targeting motor neurons.

Taken together, these findings shed light on the fact that the role played by SMN encompasses a broad spectrum of actions ranging from RNA metabolism to genome stability. However, further investigations will be needed to dissect the molecular pathway linking SMN depletion and R-loop generation and its relevance on the disease phenotype.

## 12. Alterations in Non-Coding RNAs

SMN has also been recently associated with multiple non-coding RNAs metabolism and function [138]. For example, miR-9 expression has been reported to be inversely linked with SMN protein and when the level of cellular SMN protein decreased, miR-9 levels showed a time-dependent rise [139]. Thus, miR-9 could represent an essential modulator linked to the severity of SMA. By suppressing specific critical signaling pathways in motor neurons, miR-146a produced by SMA astrocytes contributed to motor neuron dysfunction and loss [140]. Neurites of SMN-deficient neurons showed increased miR-183 expression. [141]. Finally, an increased expression of of miR-206, a muscle specific miRNA, was reported in a mouse model of intermediate SMA, suggesting it acted as a survival endogenous mechanism [142].

The SMN locus also produces a wide variety of circRNAs in addition to SMN protein. The mouse Smn locus was projected to produce a significantly lesser amount of circRNAs than the human SMN locus [143]. These variations appear to be the result of fewer short interspersed nuclear elements in the mouse Smn gene’s intronic regions. A sizable portion of the circRNAs produced from SMN were unique to primates. Therefore, it is plausible that particular circRNAs have a direct relationship with motor neuron activities that have grown more significant over time [143,144].

## 13. Conclusions

The progressive dissection of the SMN complex functions in RNA metabolism has led to the development of commercially available RNA-based therapies for SMA, which have drastically changed the outcome of this disease. However, the full picture of the multifaceted functions of SMN in post/transcriptional gene regulation and their effects on downstream cellular pathways that determine the SMA phenotype still need to be fully elucidated.

With the present review we showed the multifaced role of the SMN protein and the effect of the loss of discrete functions of SMN in the pathogenesis of SMA (Figure 1, Figure 2, Figure 3 and Figure 4). In particular, an impairment of the snRNP assembly and of the SMN complex strongly correlates with the residual levels of SMN activity in cellular models of SMA and determines downstream tissue-specific splicing alterations ([108,116,121,122,123]). The translational implications of these observations reflect in the fact that restoring snRNP function in animal models of SMA is sufficient to rescue the phenotype [109,110]. NMJ integrity as well as axonal transport, neurite and growth cones development are also affected by SMN-mediated U7 snRNP assembly/histone RNA processing and by mRNPs assembly, respectively [65,130,133,134,135,136]. Finally, the link between neuronal degeneration and the role of SMN on genome stability through the formation of R-loops is currently ongoing [85,116].

A number of interacting proteins are responsible for the achievement of all these roles of SMN, including Gemin 2-8, Unrip, Sm proteins, pICln, snRNAs, hnRNP-R, the RBP HuD, KSRP, and IMP1, β-actin mRNA, as well as 3′ UTR ARE reach transcript (i.e., Cdkn1a mRNA), RNAP II CTD and SETX, and non-coding RNAs (including miRNAs and circRNAs) (Figure 1).

Mechanistically, it will be important to define how the specificity of interactions among the protein partners and the SMN complex is achieved, and how different post/translational modifications impact on such interactions. These studies will serve as the basis to optimize current therapies, possibly leading to a multi/target treatment. In the near future, restoration of SMN obtained through RNA/based therapeutics could be complemented by SMN/independent approaches aimed at preventing motor neuron death and muscle wasting.

## Figures and Tables

**Figure 1 ijms-24-02247-f001:**
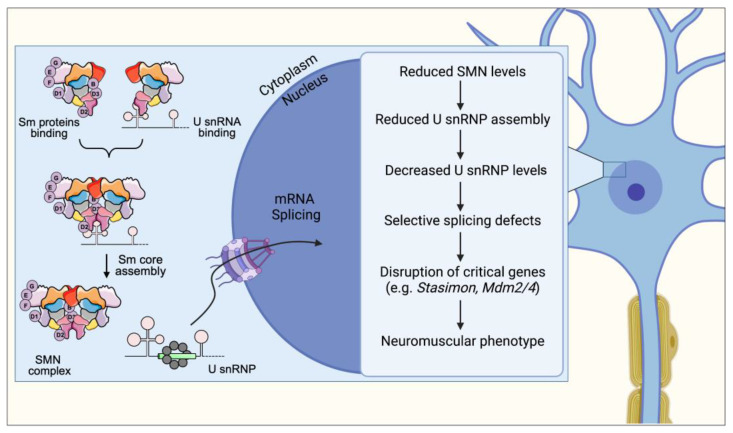
SMN is required for normal assembly and biogenesis of spliceosomal snRNPs, which are essential for proper splicing and expression of mRNAs. A cascade of molecular events links SMN deficiency to the neuromuscular phenotype in SMA. First, SMN deficiency impairs Sm core formation, leading to a decrease in snRNP levels with effects that are tissue-specific. Second, this reduction in snRNP levels causes selective splicing defects in a limited set of genes, resulting in alterations in their normal profile of expression. Third, a subset of these SMN target genes performs functions that are critical for specific neuronal classes. Lastly, disruption of the activity of these genes, such as *Stasimon*, results in selective defects of neuronal function that collectively generate the SMA phenotype.

**Figure 2 ijms-24-02247-f002:**
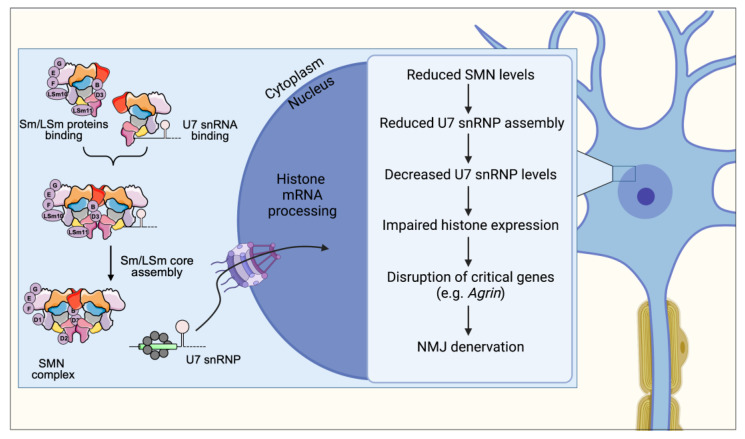
SMN controls normal assembly and biogenesis of U7 snRNP, which is required for proper 3′-end processing of histone mRNAs. In SMA, SMN deficiency disrupts U7 snRNP biogenesis leading to downstream dysregulation of histone gene expression. Altered histone expression affects the normal expression of Agrin within motor neurons leading to decreased Agrin release at vulnerable NMJs, which in turn contributes to denervation and neuromuscular pathology in SMA mice.

**Figure 3 ijms-24-02247-f003:**
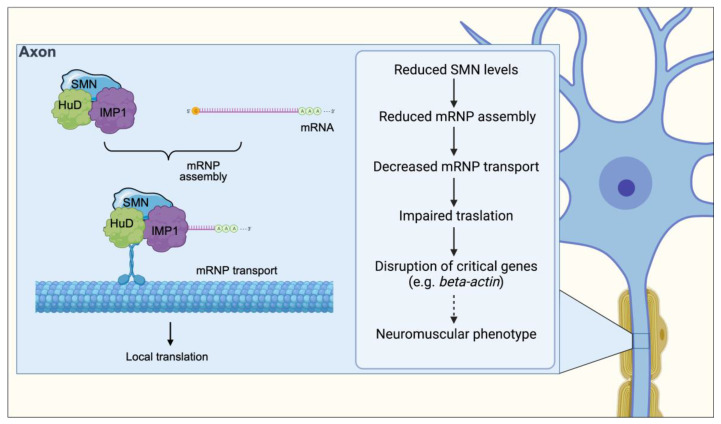
SMN facilitates the interaction of RBPs such HuD and IMP1 with their mRNA targets and contributes to neuronal mRNA trafficking. Upon being assembled in the cell body, mRNP complexes associate with motor proteins to achieve subcellular localization via both the microtubule and actin cytoskeleton for their transport along the axon. SMN deficiency causes defects in the assembly of mRNAs with RBPs leading to decreased numbers of mRNP complexes transport along the axon, resulting in a net decrease in the amount of locally translated proteins. Through a mechanism that remains to be established (dotted arrow), disruption in the expression of genes such as β-actin induces a neuromuscular phenotype.

**Figure 4 ijms-24-02247-f004:**
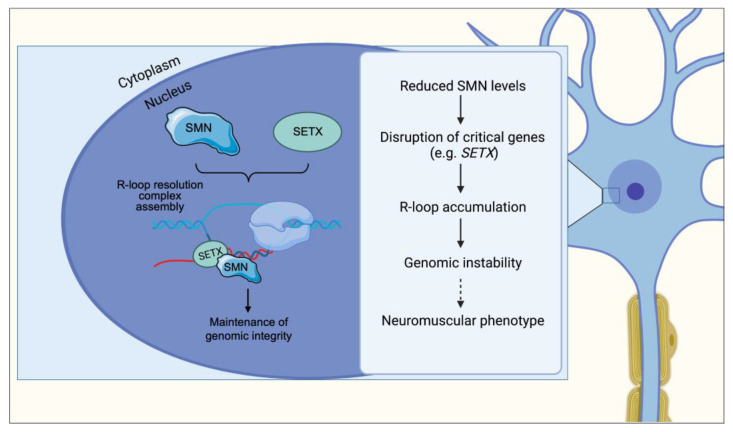
In normal cells, SMN binds to and recruits SETX onto R-loops to regulate the speed of SETX-dependent R-loop resolution. Nascent RNA hybridized to DNA is separated by the activity of R-loop-resolution complexes to make pre-mRNA available for processing and splicing. In SMA, SMN deficiency disrupts SETX expression and its interaction with R-loops, resulting in R-loop accumulation and genomic instability. Through mechanisms that remain to be established (dotted arrow), genomic instability leads to the neuromuscular phenotype observed in SMA.

## Data Availability

Not applicable.

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
