# Peer review of "The SMN Complex at the Crossroad between RNA Metabolism and Neurodegeneration"

_ijms, 2023, doi:10.3390/ijms24032247_

Round 1
Reviewer 1 Report
Dear Authors,
Your review paper “The SMN Complex at the Crossroad between RNA Metabolism and Neurodegeneration” is well written, and very interesting to a readership working in the field of neuromuscular disorders. It is methodologically correctly set, had clear goals, and the references correctly follow the entire article. However, from my point of view, a table with an overview of all the important “players” and functions will be informative for the readers. A few limitations preclude this paper from being published in its current status.
I have some recommendations:
1. What is the main update addressed by this research?
2. Which specific gap the in the knowledge of 5q- SMA do you want to address with this update?
3. What does this update add compared with other published material?
4. A discussion section is lacking!
Author Response
Response to Reviewer 1 Comments
We thank the Reviewer for the helpful comments and questions. Please, find below a point-by-point response to your requests:
Point 1: However, from my point of view, a table with an overview of all the important “players” and functions will be informative for the readers. A few limitations preclude this paper from being published in its current status.
Response 1: We thank the reviewer for the suggestions. In order to clarify the “players” and their functions described in the review, we extensively edited the figure and figure legend. After these edits we considered that an additional table will be redundant with the new information reported in the figure. However, we will be open to add an additional table if the reviewer still consider this to be required.
Reviewer: I have some recommendations:
Point 1. What is the main update addressed by this research?
Response 1: Following the reviewer suggestion, we insert a paragraph in the introduction specifying the aim of the review.
Point 2. Which specific gap the in the knowledge of 5q- SMA do you want to address with this update?
Response 2: We thank the reviewer for this valuable suggestion, our review aims to dissect the well established and recently discovered functions of the SMN complex and how they are connected to SMA pathology. We added this specification in the introduction.
Point 3. What does this update add compared with other published material?
Response 3: In our review, we aim to provide a comprehensive overview of canonical and recently discovered functions of the SMN complex, covering basic molecular concepts and adding a translational perspective.
Point 4. A discussion section is lacking!
Response 4: We thank the reviewer for the comment. In accordance to the journal format, the discussion section is named “conclusion” in our manuscript. As per the reviewer request, the conclusion/discussion section was further expanded.
Reviewer 2 Report
Faravelli et al. submitted a review on “SMN Complex at the Crossroad between RNA Metabolism and Neurodegeneration”. Overall, this manuscript is well written and provides updated literature linking RNA and SMN.
Major comments :
1) A section on non-coding RNAs such as miRNA, long noncoding RNA and circular RNA and their effect on SMN has to be added
(For example 10.1016/j.jns.2021.117485 , 10.1016/j.cellsig.2020.109696)
2) Figure 1 is difficult to follow. I suggest authors to sub label the figure 1 and re-direct to the relevant descriptions in the text. Currently I don’t see any connection between the text and figure. There is no difference in panel 2 and 3 in the figure (both in WT and SMA).
Short figure description (self-description) is needed.
Minor
Close the parenthesis in line 82
“the SMN complex (for detailed characterization see [1,3].”
Abbreviate PRMT5 at the first usage
Is it mTOR in line 154
“include TOR-dependent phosphorylation of serine 49”
Use number reference style in line 435
“in this process see Donlin-Asp et 435 al., 2016.”
Use appropriate beta symbol
“β-actin mRNA, and 436 axons from cultured SMA motor neurons exhibit reduced length and smaller growth 437 cones [63,69]. SMN is additionally important for the assembly of ß-actin mRNA with 438 Igf2-mRNA binding protein 1 (IMP1) [133]. Reduced levels of SMN are associated with 439 smaller IMP1 granules, which are less effective at binding with ß-actin mRNA”
Author Response
Response to Reviewer 2 Comments
We thank the Reviewer for the helpful comments and questions. Please, find below a point-by-point response to your requests:
Reviewer: Major comments :
Point 1) A section on non-coding RNAs such as miRNA, long noncoding RNA and circular RNA and their effect on SMN has to be added (For example 10.1016/j.jns.2021.117485 , 10.1016/j.cellsig.2020.109696)
Response 1: We thank the reviewer for this valuable suggestion, we added a section dedicated to non-coding RNAs in the manuscript.
Point 2) Figure 1 is difficult to follow. I suggest authors to sub label the figure 1 and re-direct to the relevant descriptions in the text. Currently I don’t see any connection between the text and figure. There is no difference in panel 2 and 3 in the figure (both in WT and SMA). Short figure description (self-description) is needed.
Response 2: The figure was extensively edited to better clarify each panel and make it more informative and reflect the information reported in the manuscript. The figure legend was extensively edited as well.
Reviewer: Minor
Response: The minor edits have been added to the manuscript as requested below:
Close the parenthesis in line 82 “the SMN complex (for detailed characterization see [1,3].”
Abbreviate PRMT5 at the first usage
Is it mTOR in line 154 “include TOR-dependent phosphorylation of serine 49”
Use number reference style in line 435 “in this process see Donlin-Asp et 435 al., 2016.”
Use appropriate beta symbol
“β-actin mRNA, and 436 axons from cultured SMA motor neurons exhibit reduced length and smaller growth 437 cones [63,69]. SMN is additionally important for the assembly of ß-actin mRNA with 438 Igf2-mRNA binding protein 1 (IMP1) [133]. Reduced levels of SMN are associated with 439 smaller IMP1 granules, which are less effective at binding with ß-actin mRNA”
Reviewer 3 Report
Very nice paper, reads well and the story is easy to follow. One minor change, in lines 37 and 40 you talk about non-coding RNAs but you used different spelling - could you pick one for both lines?

Author Response
Response to Reviewer 3 Comments
Point 1: Very nice paper, reads well and the story is easy to follow. One minor change, in lines 37 and 40 you talk about non-coding RNAs but you used different spelling - could you pick one for both lines?
Response 1: We thank the Reviewer for the comment about our manuscript.
Spelling was edited as requested in line 37 and 40 (non-coding RNA).